# Differential Expression of Kinin Receptors in Human Wet and Dry Age-Related Macular Degeneration Retinae

**DOI:** 10.3390/ph13060130

**Published:** 2020-06-24

**Authors:** Rahmeh Othman, Simon Berbari, Elvire Vaucher, Réjean Couture

**Affiliations:** 1School of Optometry, Université de Montréal, Montreal, QC H3T 1P1, Canada; Rahmeh.othman@umontreal.ca (R.O.); Elvire.vaucher@umontreal.ca (E.V.); 2Department of Pharmacology and Physiology, Faculty of Medicine, Université de Montréal, Montreal, QC H3T 1P1, Canada; Simon.berbari@umontreal.ca

**Keywords:** age-related macular degeneration, bradykinin type 1 receptor, fibrosis, gliosis, iNOS, kallikrein-kinin system, microglia, vascular endothelial growth factor (VEGF), retina

## Abstract

Kinins are vasoactive peptides and mediators of inflammation, which signal through two G protein-coupled receptors, B1 and B2 receptors (B1R, B2R). Recent pre-clinical findings suggest a primary role for B1R in a rat model of wet age-related macular degeneration (AMD). The aim of the present study was to investigate whether kinin receptors are differentially expressed in human wet and dry AMD retinae. The cellular distribution of B1R and B2R was examined by immunofluorescence and in situ hybridization in post-mortem human AMD retinae. The association of B1R with inflammatory proteins (inducible nitric oxide synthase (iNOS) and vascular endothelial growth factor A (VEGFA)), fibrosis markers and glial cells was also studied. While B2R mRNA and protein expression was not affected by AMD, a significant increase of B1R mRNA and immunoreactivity was measured in wet AMD retinae when compared to control and dry AMD retinae. B1R was expressed by Müller cells, astrocytes, microglia and endothelial/vascular smooth muscle cells, and colocalized with iNOS and fibrosis markers, but not with VEGFA. In conclusion, the induction and upregulation of the pro-inflammatory and pro-fibrotic kinin B1R in human wet AMD retinae support previous pre-clinical studies and provide a clinical proof-of-concept that B1R represents an attractive therapeutic target worth exploring in this retinal disease.

## 1. Introduction

Age-related macular degeneration (AMD) features neurodegeneration of the retina, principally the macula, leading to a loss of central vision. The global prevalence of AMD increases dramatically with aging, and it is estimated to affect 192 million people over the world in 2020 [1]. In its early stage, AMD is asymptomatic, although lipid deposits known as drusen accumulate in the subretinal space. These deposits consequently affect the retina, leading to a thinning in the photoreceptor layer [2] and a loss of sensitivity, especially in the macula. AMD can progress with time evolving into two distinct late stages: geographic atrophy also known as ‘dry AMD’ and neovascular or ‘wet AMD’ [3]. Both forms are linked to inflammation and dysregulated innate immunity [4]. Although the reasons behind the progression of AMD into the late stage remain elusive, many mechanisms seem to take part in the progression, notably the complement’s activation, oxidative stress, inflammation, blood flow dysregulation and altered retinal neuroprotection [5].

Dry AMD is typified by a geographic atrophy consisting of a thickening in Bruch’s membrane (BrM) in the macula, and accumulation of microglia in the subretinal space, as well as accumulation of macrophages, a degeneration of the retinal pigment epithelium (RPE) and overlying photoreceptors (PR), accompanied by choroidal blood vessels constriction [6,7]. The wet form of AMD is characterized by the ingrowth of new and immature vessels, which invade the outer retina from the underlying choroid through a defect in Bruch’s membrane. This choroidal neovascularization (CNV) creates the obstruction of light paths and possible retinal detachment leading to impaired vision. In both forms, the locus of the primary insult remains elusive [6,8]. A number of studies debate whether the dry form (90% of AMD) can evolve into the wet form, or whether they represent two different pathologies [9]. Current clinical trials are designed to answer this question, in order to find adequate therapeutic treatments for these retinal diseases [10]. The common treatment of wet AMD targets the overexpressed vascular endothelial growth factor A (VEGFA), implicated in CNV formation, by means of the intravitreal injection of anti-VEGF antibodies [6,11,12,13,14]. There is no specific treatment for dry AMD, except for dietary and supplementary vitamins and antioxidants, which have not been proven efficient [5,10].

Recent studies support a role for the kallikrein-kinin system (KKS) in retinal damage, notably in diabetic retinopathy and CNV in animals and humans [15,16,17,18,19]. A Phase 1B clinical trial was recently conducted targeting plasma kallikrein, the enzyme involved in the biosynthesis of kinins, to treat diabetic macular edema. In this study, intravitreal injection of KVD001 (plasma kallikrein inhibitor) ameliorated visual acuity without causing any retinal thickening or exacerbating the pathology [20]. Collectively, these pre- and clinical findings support a key role for the KKS in retinal disorders. The vascular biology and pathology of the KKS, including inflammation and fibrosis, operate through bradykinin B1 and B2 receptors (B1R and B2R) [21,22,23,24]. The constitutive B2R plays an essential role in vascular homeostasis [24], but also in tumor-induced angiogenesis and pathological neovascularization [25,26]. B2R blockade reduced retinal neovascularization and inhibited the expression of proangiogenic and pro-inflammatory cytokines in oxygen-induced retinopathy [25]. These mechanisms could be linked to VEGF cascade, since B2R transactivates VEGF type 2 receptor (VEGFR2) (KDR/Flk-1), resulting in further vasodilatation, microvascular permeability and angiogenesis [27]. Most of the detrimental effects of the KKS are associated to the inducible B1R [24]. B1R shows deleterious effects in the early and late stage of type 1 diabetic retinopathy via the inducible nitric oxide synthase (iNOS) pathway [15,17,18], and in laser-induced CNV via pro-inflammatory cytokines, microglia invasion and astrogliosis [16]. Importantly, B1R antagonism and anti-VEGF therapy decreased the size of CNV, retinal inflammation and vascular hyperpermeability through distinct and complementary mechanisms [16].

Based on recent and compelling pre-clinical findings suggesting a primary role for kinin B1R in CNV, the aim of this study was to address the hypothesis that inflammatory events associated to kinin B1R also occur in human AMD retina. Post-mortem human dry and wet AMD retinae were used to study the differential expression and cellular distribution of kinin B1R and B2R by immunofluorescence and in situ hybridization. Furthermore, B1R was studied in association with inflammatory proteins (iNOS and VEGFA), fibrosis markers, reactive microglia and gliosis. Hence, a better understanding of the cellular inflammatory mechanisms underlying both forms of AMD should be useful in the search for new therapies.

## 2. Results

### 2.1. B1R, B2R and VEGFA RNA Expression

To assess the implication of the inducible B1R and constitutive B2R and VEGFA, we first looked at the mRNA expression of these receptors and proteins in control and AMD retinae by Multiplex RNAscope (Figure 1). Interestingly, clusters of B1R mRNA were observed in the ganglion cell layer (GCL), inner nuclear layer (INL) and outer nuclear layer (ONL) of dry and wet AMD (Figure 1F,J), but not in the control retina (Figure 1B). B1R mRNA expression in wet AMD (Figure 1J) was significantly different (GCL and INL) from the control retina (Figure 1B,M), but not from the dry form (Figure 1F,M). B1R mRNA expression was significantly higher only in GCL in the dry form, in comparison to the control retina (Figure 1M). B2R mRNA expression was quite low and not significantly different between the two forms of AMD (Figure 1G,K,N) or in comparison with the control retina (Figure 1C,N), yet most of the microvessel walls highly expressed B2R mRNA (not quantified). VEGFA mRNA expression was striking in each retina, especially in the GCL and INL. Surprisingly, compared to the control retina (Figure 1D,O), VEGFA mRNA expression in the INL was significantly lower in the wet AMD retinae, but it was similar in the other layers (Figure 1L,O). The reason for this low mRNA expression in the wet AMD (INL) has not been investigated; one hypothesis that can be made is that this change is caused by a medication taken by these patients. For example, it was reported that statins, used in most elderly patients, decrease VEGF expression by reducing the oxidative stress in RPE culture [28]. Unfortunately, the use of medication was not reported in the medical history of the donors used in this study, although we can confirm that they did not receive any anti-VEGF therapy. Some unidentified structures expressed the three markers, but it was not clear whether this was degenerating retinal ganglion cells or lipofuscin deposits.

### 2.2. B1R Immunoreactivity and Cellular Distribution

B1R immunoreactivity was scarcely detectable in control retinae (Figure 2, left panel). In the dry AMD, the level of B1R staining in all layers (GCL, inner plexiform layer (IPL), INL, outer plexiform layer (OPL), ONL) tended to increase compared to control, but this did not achieve statistical significance (Figure 2A, middle panel, and Figure 2B). In contrast, B1R staining was significantly stronger in GCL, IPL, INL and ONL of the wet AMD retinae, when compared to the control counterparts (Figure 2A, right panel, and Figure 2B). B1R staining was also intense in the photoreceptor layer of the wet AMD retinae, however, this layer was often damaged and the high level of autofluorescence in the external segments of PR prevented accurate quantification.

Further investigation was performed to determine the cell-type expression of B1R immunoreactivity, particularly on retinal vessels. B1R staining was absent in control blood vessels, but it was intense in the AMD retinae, especially in the wet form, where B1R was expressed in smooth muscle and endothelial cells (Figure 3). Moreover, B1R immunoreactivity was found in the RPE of the dry and more intensively of the wet AMD retinae, but was absent in the RPE of control retinae (Figure 3). Due to the fragility of the pathological RPE layer, RPE was not present in all sections. Thus, B1R observation in RPE was limited to two patients in the wet form, three in the dry AMD and four in the controls.

### 2.3. Macroglia Immunoreactivity

We next evaluated the reactivity of the retinal macroglia in the two forms of AMD, and their expression of B1R. Macroglia, i.e., astrocytes and Müller cells, as a population of retinal cells, become activated in the presence of cellular stress and in aging to protect the retina and overexpress the glial fibrillary acidic protein (GFAP) [29,30]. However, a chronic activation of these cells accelerates the evolution of the pathology, opens the blood-retinal barrier, which facilitates the infiltration of toxic compounds and participates to neovascularization [29]. Thus, the immunoreactivity of macroglia cells was observed using the GFAP marker (Figure 4).

In immunofluorescence staining retinal sections, a slight but not significant increase of GFAP expression was observed in the dry AMD (Figure 4F,J), when compared to the control (Figure 4C,J). However, in the wet AMD retinae, an intense reactivity of GFAP was observed (Figure 4I,J). The GFAP immunoreactivity was significantly increased in the IPL, INL, OPL and ONL. Interestingly, a partial colocalization of B1R with GFAP was observed in the wet AMD mostly in the outer retina (Figure 4G).

### 2.4. Microglia Immunoreactivity

One of the hallmarks of retinal degeneration is microglia activation [31]. Hence, we assessed microglia cell expression, using ionized calcium binding adaptor molecule 1 (Iba1) marker (Figure 5). Iba1 expression was restricted to the inner retina (GCL and INL) in the control and dry AMD retinae. In control retina, microglia has small soma with thin ramifications, whereas in AMD retinae microglia activation was observed, and it was indicated by two phenotypes: hypertrophic somas and retraction of ramifications, and an amoeboid shape indicating a fully activated microglia [4]. The amoeboid shape was highly present in the wet AMD, especially in the outer layers, RPE and ONL (Figure 5I,L,N). Moreover, microglia cell proliferation (number of microglia/100 µm^2^) of the amoeboid shape was noticed in the wet AMD retinae, but not in the dry AMD (Figure 5O). Double immunostaining of Iba1 and B1R showed a colocalization of these two proteins only in the wet AMD retinae (Figure 5G,J).

### 2.5. iNOS Immunoreactivity

We then assessed the expression of iNOS, an enzyme contributing to increase the oxidative stress and by which B1R mediates inflammation in diabetic retinopathy [17]. The levels of iNOS in the dry AMD retinae (Figure 6F) were similar to the levels of the control retinae (Figure 6C) in all retinal layers (Figure 6J). In contrast, iNOS expression was significantly increased in all layers of the wet AMD retinae in comparison with the dry AMD and control retinae (Figure 6I,J). Moreover, the double labeling of B1R and iNOS showed a colocalization of these two proteins in all layers of the wet AMD retinae (Figure 6G), but not in the control (Figure 6A) or dry AMD retinae (Figure 6D).

### 2.6. Fibrosis Formation

Since fibrotic lesions was shown in neovascular AMD [32], and fibrosis was associated with an upregulation of B1R [33], we evaluated the occurrence of fibrosis in both forms of AMD using the expression of α-SMA and collagen 1α as markers [34,35]. Neither the control (Figure 7C,D) nor the dry AMD (Figure 7G,H) retinae showed any specific staining to evidence fibrosis formation. However, an increase of immunoreactivity of α-SMA and collagen 1α was seen in the wet AMD retinae, particularly in the outer layers (Figure 7K,L). More interestingly, a colocalization of collagen 1α and α-SMA with B1R was observed in all the outer layers (OPL and ONL) of the wet AMD (Figure 7I), but not in the dry AMD (Figure 7E) or control (Figure 7A) retinae.

### 2.7. B2R Immunoreactivity

Since B2R genomic expression was unchanged in the AMD retinae (Figure 1), we checked whether this can be translated at protein level in AMD retinae. Immunofluorescence staining failed to show any significant changes in the expression of B2R in the dry and wet AMD in comparison with the control retinae (Figure 8A,B). No further investigation was made to identify the cell-type bearing the B2R in human retinae.

## 3. Discussion

The present study provides the first demonstration that B1R is highly expressed in human wet AMD retina, but weakly in the dry form. In the wet AMD retinae, B1R was particularly densely expressed in blood vessels and RPE, which suggests an implication of the inducible B1R in the alteration of blood-retinal barrier and the outer retinal-barrier in AMD. Moreover, B1R was colocalized with iNOS, microglia and macroglia markers, as well as with fibrosis markers throughout the wet AMD retina, supporting a role for B1R in the inflammatory process in AMD. This is in keeping with our team’s earlier data showing that B1R is overexpressed in laser-induced CNV in rodents, where B1R blockade had beneficial effects on both inflammation and neovascularization [16]. Hence, these findings support an implication of the B1R in wet AMD, while its involvement in the dry form remains uncertain on the basis of its weak immunocytochemical expression. One potential important outcome arising from this research is the possibility to treat wet AMD with an antagonist of B1R to reduce inflammation and fibrosis. In contrast, B2R expression was not altered in the dry and wet AMD retinae, highlighting a differential expression of both kinin receptors in AMD.

### 3.1. B1R in Human Wet AMD

An upregulation of B1R expression in a rat model of CNV was recently reported [16]. Moreover, several studies suggested a major implication of the B1R in retinal pathologies, by increasing the inflammatory response, vascular permeability and leukostasis [15,17,18]. Here, we show an upregulation of B1R in all layers of AMD retinae, with a more prominent expression in the wet form. Interestingly, our histopathologic study shows an expression of B1R on the endothelial cells of the retinal blood vessels of wet AMD retinae. This is in agreement with our previous study, showing B1R expression on retinal endothelial cells in rat CNV [16]. Endothelial cell dysfunction is a major contributor to the opening of the blood-retinal barrier in retina including diabetic retinopathy and AMD [29]. The presence of B1R on endothelial cells of the human AMD retina may contribute to endothelial dysfunction and blood-retinal barrier alteration. B1R expression was also expressed on vascular smooth muscle cells in the wet AMD retinae. With few exceptions, B1R activation constricts veins and dilates arteries through endothelium/NO pathway [36,37]. B1R arterial/arteriolar-induced dilation associated with post-capillary constriction would lead to increased capillary hydrostatic pressure and subsequent plasma protein extravasation and oedema. This is supported by several studies showing a major contribution of B1R in enhanced retinal vascular permeability, which was reversed by B1R blockade [16,17,18,38]. While B1R can also affect the retinal blood flow, it is still uncertain whether B1R can contribute to retinal ischemia through a prolonged stimulatory action on vascular smooth muscle cells. Retinal ischemia is a major contributor of neovascularization through VEGFA activation [29]. B1R is implicated in retinal neovascularization in laser-induced CNV through a mechanism independent of the VEGFA/VEGFR2 axis, and B1R antagonism reduced CNV [16]. This is reminiscent of the neovascularization induced by the B1R agonist (Lys-des-Arg^9^-BK), which was sensitive to B1R antagonism in the rabbit cornea [39]. Collectively, these data suggest an implication of B1R in neovascularization.

Interestingly, B1R was highly expressed in the outer blood-retinal barrier and RPE of the wet AMD retinae. The RPE layer separates the retina from the choroid, providing a protection against outer insults. Hence, an alteration of this layer exposes the retina to the choroidal circulation and to an infiltration of pro-inflammatory immunomodulators. Choroidal neovascularization in patients with AMD induces the loss of the outer blood-retinal barrier by down-expression of specific genes like RLBP1, RPE65 and VEGFA [40]. Despite the important role of the choroid and Bruch’s membrane in the development of AMD pathology, the RPE is considered as a fulcrum of AMD pathogenesis [9]. Indeed, RPE dysfunction precedes the end stages of the wet and dry AMD [8]. Therefore, the high expression of B1R in the RPE of human AMD retinae points towards an implication of this receptor in the alterations caused to the RPE layer. Indeed, inflammation and oxidative stress are the main participants in the alteration of RPE in AMD. Targeting one of these two events can slow down and even prevent RPE degeneration [8,41]. It is worth noting that B1R was expressed in the RPE and perpetuated inflammation and oxidative stress in diabetic retinae [15,17,18]. Hence, the presence of B1R in the RPE may contribute to RPE degeneration through the inflammation and oxidative stress in human AMD.

### 3.2. Glia Cells in AMD

The mammalian glia includes macroglia (astrocytes, Müller cells) and microglia. Glial cells are dynamic keepers of the tissue and become activated following an insult to the retina [4,6]. In the present study, an increase in GFAP immunoreactivity, considered a hallmark of macro-gliosis [42], was observed in all layers of the wet AMD retinae and mainly in the GCL of the dry AMD retinae, indicating an increase in astrocyte reactivity. Müller cells have similar roles as astrocytes, and they span across the retina [43]. Although activated macroglia provides retinal neuroprotection, their chronic activation is harmful for the retina [44]. Chronic gliosis increases vascular permeability and neovascularization in retina, thus exacerbating the disease progression [29]. Moreover, our study shows a co-expression of GFAP with the B1R in human AMD retinae. This is in line with our previous results in a pre-clinical model of AMD [16]. Furthermore, B1R was shown to be expressed on brain astroglia in mice models of encephalomyelitis and Alzheimer’s disease [45,46], and blocking B1R had beneficial effects on the pathology of Alzheimer’s [46]. Inhibition of B1R also protects from brain injury by reducing astroglia activation in mice [47]. Hence, our observation about the colocalization of B1R with GFAP suggests an implication of B1R in chronic gliosis in human AMD.

Macroglia interact with microglia in health and retinal disease [48], and regulate its activation and phagocytic function [49]. Microglia are resident macrophages, playing an essential role in protecting retina against noxious insults. With aging, microglia display increased signs of gliosis indicated by a retraction of ramifications and soma hypertrophy, and the amoeboid phenotype prevails when microglia are highly activated [4]. Our study shows a more prominent expression of Iba1, a microglia marker, in the wet than in the dry AMD. However, in both forms, activated microglia (amoeboid) were more expressed than the quiescent microglia (ramified). A high expression of amoeboid microglia was observed in the outer retina of the wet AMD, especially between the RPE and Bruch’s membrane. These observations suggest an infiltration of microglia that occurs in the wet AMD. In contrast, activated microglia in the outer retina of the dry AMD was less observed. This is consistent with the less impressive role of microglia in the dry AMD [6] and their harmful role in the wet form, contributing to the progression of the pathology and the degeneration of photoreceptors [4]. Moreover, our immunofluorescence staining showed a co-expression of B1R with the amoeboid, but not with the ramified microglia all over the wet AMD retinae, particularly in the outer layers of the wet AMD (ONL and RPE). This is reminiscent of the colocalization of B1R with Iba1 in our rat model of laser-induced CNV [16]. These observations suggest an implication of B1R in microglia activation. Moreover, the high B1R co-expression with microglia in the outer retina points to its implication in microglia infiltration in AMD. Indeed, in cultured microglia, the B1R, but not the B2R, agonist induced microglia migration, motility and chemotaxis, and this response was blocked by B1R antagonists [50].

### 3.3. Oxidative Stress in AMD

Several studies highlighted the contribution of oxidative stress in AMD progression, where plasmatic and retinal levels of oxidative stress markers were found increased in patients with AMD [51,52,53]. Moreover, recent studies showed that increased oxidative stress in RPE cells alters not only coding genes, but also non-coding genes related to biochemical pathways associated with all major fields of cellular metabolism [54,55,56]. Here, we show an upregulation of iNOS expression in the human wet AMD, but not in the dry form. Moreover, iNOS was co-expressed with the B1R in the wet AMD retinae. This is in line with previous observation showing that B1R increases oxidative stress and activates iNOS in diabetic retinopathy [17], and blocking B1R inhibited iNOS upregulation and oxidative stress [18]. B1R through iNOS and NADPH oxidase activation leads to peroxynitrite formation, a highly cytotoxic molecule that causes DNA damage, lipid peroxidation, proteins nitration, inflammation and major damage to vessels [57]. In conformity with this, changes of blood flow parameters related to vessels damage in human glaucoma were attributed to peroxynitrite formation [58]. One of the current strategies to treat AMD is to target oxidative stress. Hence, a major clinical trial (Age-Related Eye Disease Study (AREDS)) showed that dietary and supplementary anti-oxidants intake (vitamin C, vitamin E, beta-carotene and zinc) decreased the risk of advanced AMD and its associated vision loss [59].

### 3.4. Fibrosis in AMD

Despite improving visual acuity, unsuccessful treatments with anti-VEGF in AMD have often been attributed to the formation of subretinal fibrosis [60]. The progression of photoreceptor destruction was found to be proportional to the diameter and thickness of the subretinal fibrosis [61]. Indeed, subretinal fibrosis can lead to degeneration of RPE, photoreceptors and choroidal vessels [60]. Interestingly, our histopathologic studies have shown an intense immunoreactivity of fibrosis markers (collagen 1α and α-SMA) in the wet AMD, but not in the dry form, co-localized with the B1R. Several studies showed a contribution of B1R in fibrosis formation. For instance, in diabetic cardiomyopathy, B1R mediated inflammation and fibrosis [22]. In human embryonic lung fibroblasts, B1R activation induced collagen I synthesis and fibrogenesis [62]. B1R blockade reversed inflammation and the expression of fibrosis markers (collagen and α-SMA) in renal fibrosis [23,33]. Here, we show a colocalization of B1R with the fibrosis markers (collagen 1α and α-SMA), suggesting a possible implication of B1R in fibrosis formation in human wet AMD.

### 3.5. B1R and B2R in Neovascularization

B2R plays a key role in physiological and less commonly in pathological neovascularization [24]. Indeed, B2R blockade in laser-induced CNV in mice decreased VEGF expression [63]. B2R partakes in tumor angiogenesis [26] and in pathological neovascularization in oxygen-induced retinopathy, where B2R blockade reduced retinal neovascularization and the expression of proangiogenic and pro-inflammatory cytokines [25]. However, our immunofluorescence and mRNA assay did not show any changes in B2R expression in the wet and dry AMD retina. Thus, the role of B2R in AMD remains doubtful, particularly in absence of changes in RNA and protein expression.

The role of B1R in angiogenesis was first reported in 1993 [64]. In a model of limb ischemia, B1R was shown to promote neovascularization [65], and this response was reduced in B1R knocked-out mice [66]. In this study, we found a weak induction and expression of B1R in the dry AMD, in comparison with the prominent upregulation of B1R in the wet form, suggesting a possible role of B1R in the neovascularization events of wet AMD. This hypothesis is supported by the pre-clinical finding showing that B1R blockade reversed the inflammatory response and the CNV size in rat retina [16]. In the latter study, B1R was found to partake to the inflammatory process in CNV independently of the VEGFA pathway [16].

## 4. Materials and Methods

### 4.1. Subjects

Post-mortem human eyes were obtained from the Human Eye Biobank for Research at St. Michael’s Hospital, Toronto, Canada. The retina donation and the protocols were conformed to the 2013 Helsinski declaration and the ethical standards of the Ethical Committee of Health Research, Université de Montréal (17-142-CERES) and St. Michael’s Hospital Research Ethics Board. Donors signed a consent donation form when they were alive, or the assent of a parent was obtained after death. Eyes from five patients with dry AMD (83 ± 4 years), five patients with wet AMD (92 ± 4 years) and five controls (71 ± 5 years) were studied (Table 1). Subjects were excluded if post-mortem time-to-fixation was greater than 24 h, and subject information reports indicated that none of the patients received any anti-VEGF treatment for AMD.

### 4.2. Immunofluorescence

B1R, B2R, iNOS, fibrosis markers (α-smooth muscle actin (α-SMA) and collagen1 α) and glial cells markers (glial fibrillary acidic protein (GFAP) for macroglia and ionized calcium-binding adapter molecule 1 (Iba1) for microglia) were detected by immunohistochemistry on paraffin embedded retinae, as described previously [17]. Sections obtained from the Eye Biobank were deparaffinized prior to the immunofluorescence procedure. Glass slides were first incubated in sodium citrate buffer at 95 °C for 35 min. Sections were then left to cool down for 20 min at room temperature (RT), then washed three times (3 × 5 min) with 0.1 M PBS buffer (pH 7.4) and incubated for 1 h at RT in blocking buffer (PBS containing 10% donkey serum or goat serum and 0.25% triton X-100). Sections were left incubated overnight at RT, with the blocking buffer containing primary antibodies (Table 2).

The following day, slides were washed 3 × 5 min in PBS 0.1M and then incubated for 2 h at RT with secondary antibodies: Alexa Fluor 555 goat anti-rabbit (1:200, A21428), Alexa Fluor 488 goat anti-rabbit (1:200, A21206), Alexa Fluor 555 donkey anti-mouse (1:200, A31570), Alexa Fluor 488 donkey anti-mouse (1:200, A21202), Alexa Fluor 546 donkey anti-goat (1:200, A11056), Alexa Fluor 555 goat anti-chicken (1:200, A21437), Alexa Fluor 633 donkey anti-goat (1:200, A21082), all purchased from Life Technologies, Burlington, ON, Canada. To eliminate the autofluorescence caused by tissue components and by the accumulation of lipofuscin [68], an autofluorescence eliminator reagent (2160, Sigma Aldrich, Oakville, ON, Canada) was used. The slides were then washed and mounted using a homemade glycerol solution. Images were obtained with a confocal microscope Zeiss-LSM800 equipped with an argon laser (Carl Zeiss, Jena, Germany) and transferred to a computer and analyzed using NIH ImageJ 1.36b Software (NIH, Bethesda, MD, USA). Images were obtained at 40× and 60× objectives. Semi-quantification of immunofluorescence staining intensity was made on five randomly selected surface areas of each retina from five dry AMD, five wet AMD and five controls. Background intensity (gray intensity) was subtracted from each individual value.

### 4.3. In Situ Hybridization: RNA Scope Assay

To visualize and quantify the genomic expression of B1R, B2R and VEGFA, RNA scope fluorescent multiplex reagent kit (Advanced Cell Diagnostics (ACD), Newark, IL, USA) was used, following the manufacturer recommendations. First, tissue sections were deparaffinized with xylene (2 × 5 min) and alcohol 100% (2 × 5 min). To permeabilize cells and unmask target RNA, sections were pre-treated with H_2_O_2_ for 10 min at RT, washed with distilled water for 2 min, and incubated for 25 min at 95 °C in target retrieval 1× solution (Cat no. 322335). Slides were then washed for 15 s with distilled water, prior to a wash for 3 min with 100% alcohol. Sections were left to dry out for 5 min at RT. To break the nuclear membrane, sections were treated with a protease III solution (Cat no. 322337) at 40 °C for 30 min and then washed twice for 5 min each with distilled water. To target RNA molecules, sections were hybridized with a mix of three probes (Hs-BDKRB1, Hs-BDKRB2-C2, Hs-VEGFA-C3 for B1R, B2R and VEGFA, respectively). Three amplifiers for each probe (respectively AMP1, AMP2 and AMP3) were added consecutively, and incubated each time at 40 °C for 30 min. Sections were washed using wash-buffer and the signals were developed with horseradish peroxidase HRP-C signal Opal specific for each probe (Opal 520 for B1R detection, Opal 570 for VEGFA detection and Opal 690 for B2R detection) and incubated at 40 °C for 15 min, followed by another incubation at 40 °C for 30 min. The reactions were stopped by incubating a section with HRP blocker at 40 °C for 15 min. Sections were counterstained with 4′,6-diamidino-2-phenylindole (DAPI) and mounted using a homemade glycerol solution. Images were obtained with a confocal microscope Zeiss-LSM800 equipped with an argon laser (Carl Zeiss, Jena, Germany) and transferred to a computer and analyzed using NIH ImageJ 1.36b Software (NIH, Bethesda, MD, USA). Images were obtained at 40× objective. The quantification of fluorescence staining intensity was made using ACD guidelines. Briefly, five microphotographs (160 µm^2^ of dimensions) were randomly taken for each retina from five dry AMD, five wet AMD and five controls. Each fluorophore channel was analyzed separately, using each nuclear layer as a region of interest. At least twenty dots were selected in each region, the area of the integrated intensity of each dot was then measured. The average intensity per single dot was then calculated using the following formula: (average of integrated intensity of selected dots − average background intensity · average area of selected dots)/number of selected dots). The total dot intensity in the region of interest was then calculated using the following formula ((total intensity of region of interest − Average background intensity · total area)/average intensity per single dot). The average of intensity per cell was then counted by dividing total immunofluorescence intensity in the chosen region of interest by the total number of cells in the same region.

### 4.4. Statistical Analysis of Data

Results are expressed as the mean ± SEM, and n represents the number of patients. Statistical analysis of data was performed using Prism^TM^ version 5.0 (GraphPad Software Inc., La Jolla, CA, USA). Kruskal-Wallis, followed by a Dunn post hoc test, was used for comparison between the control, dry AMD and wet AMD. The results were considered significant at *p* < 0.05.

## 5. Conclusions

The use of post-mortem human retinae in this study provided an opportunity to expand our knowledge of wet and dry AMD pathology. A summary of the key findings is provided in Table 3. The limitations of this study include variation between subject ages, and the lack of a detailed AMD and medical history on each donor eye. However, none of those patients have received any treatment for AMD, and our results confirmed an intense immunoreactivity for B1R in all the wet AMD retinae, in comparison with a weak expression in the dry AMD retinae and nearly no expression in the control retinae. Importantly, B1R was co-expressed with several inflammatory/fibrosis markers and, notably, glial cells in the wet form only. Thus, it was obvious that inflammatory events are exacerbated in the wet form compared to the dry form, especially with regard to B1R expression, reactive gliosis and microglia activation and infiltration. While more studies are needed to determine the precise role of B1R in AMD, the induction and upregulation of this pro-inflammatory and pro-fibrotic receptor in human AMD retina support previous pre-clinical studies, and provide a clinical proof-of-concept that B1R represents an attractive therapeutic target worth exploring in AMD.

## Figures and Tables

**Figure 1 pharmaceuticals-13-00130-f001:**
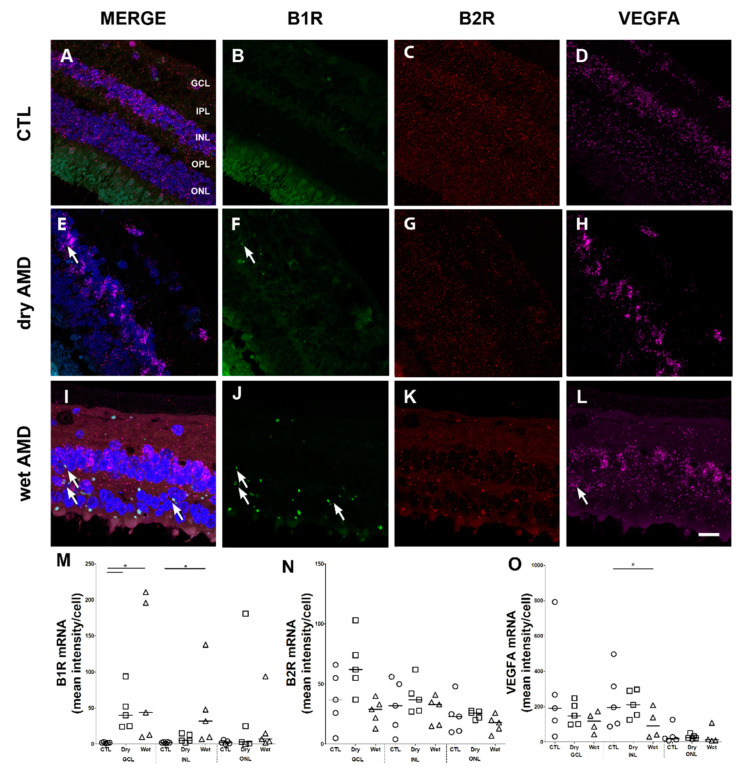
B1 and B2 receptors (B1R, B2R) and vascular endothelial growth factor A (VEGFA) mRNA expression in human retinae (**A**–**L**). Microphotographs showing the mRNA labeling for B1R (green; **B**,**F**,**J**), B2R (red; **C**,**G**,**K**) and VEGFA (purple; **D**,**H**,**L**) in the control (CTL), dry age-related macular degeneration (AMD) and wet AMD retinae. Sections are counterstained for 4′,6-diamidino-2-phenylindole (DAPI; blue; **A**,**E**,**I**), which labels cell nuclei. Note that B1R was partially colocalized with B2R in wet AMD in the outer nuclear layer (ONL), and B2R was also partially colocalized with VEGFA in the ONL, however, there was rare colocalization between B1R and VEGFA. Images were obtained at 40×. Scale bar: 20 μm. (**M**–**O**) Quantification of the mRNA (mean fluorescence intensity/cell) of B1R, B2R and VEGFA in the different layers of the retina. Data are mean ± SEM of values obtained from five retinae per group and four zones per retina. * *p* < 0.05 AMD compared with control. CTL: control (O), Dry: dry AMD (□), Wet: wet AMD (△), GCL: ganglion cell layer, IPL: inner plexiform layer, INL: inner nuclear layer, OPL: outer plexiform layer, ONL: outer nuclear layer.

**Figure 2 pharmaceuticals-13-00130-f002:**
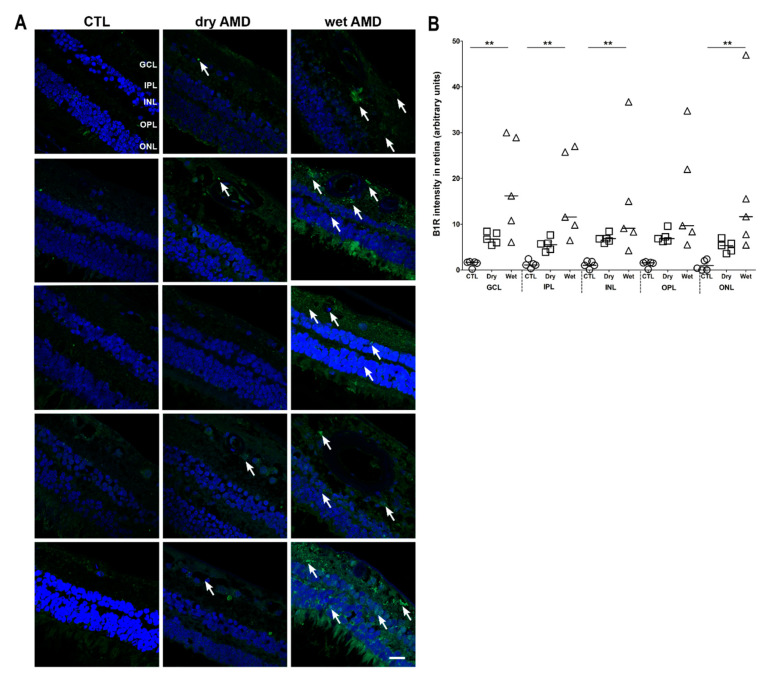
B1R immunoreactivity in human retinae. (**A**) Microphotographs show B1R in five human control (left panel), five dry (middle panel) and five wet (right panel) AMD retinae. B1R (green) was not detected in control retina, and mostly expressed in the GCL of the dry AMD retinae (arrows). In the wet AMD, B1R was highly detected in most retinal layers. Sections are all counterstained for 4′,6-diamidino-2-phenylindole (DAPI; blue), which labels cell nuclei. Images were obtained at 40×. Scale bar: 20 μm. (**B**) Semi-quantification of B1R fluorescence intensity on all the surface area of each retinal layer from five dry AMD, five wet AMD and five controls showing a significant increase (** *p* < 0.005) of B1R labeling intensity in the wet AMD, compared to the control retina. Data represent mean intensity ± SEM after subtraction of the background intensity. CTL: control (O), Dry: dry AMD (□), Wet: wet AMD (△), GCL: ganglion cell layer, IPL: inner plexiform layer, INL: inner nuclear layer, OPL: outer plexiform layer, ONL: outer nuclear layer.

**Figure 3 pharmaceuticals-13-00130-f003:**
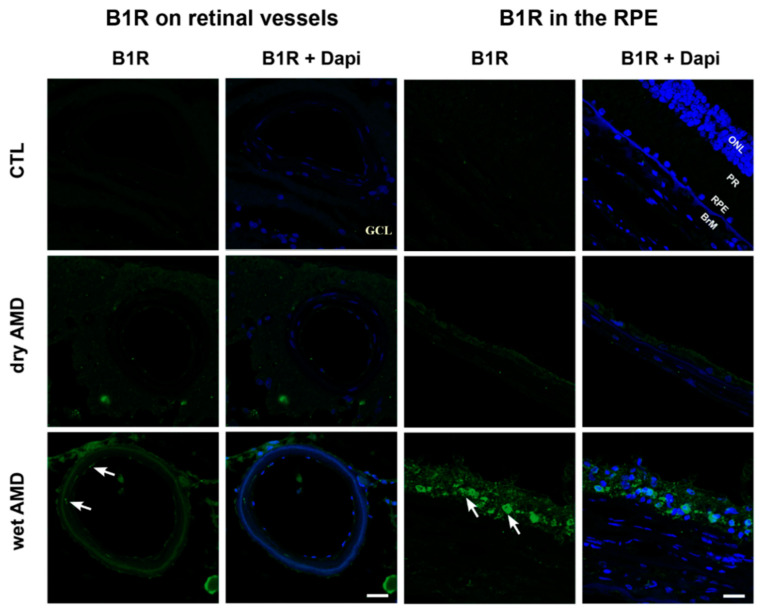
Microphotographs of immunolocalization of B1R in smooth muscle and endothelial cells of blood vessels (arrows) and in the retinal pigment epithelium (RPE) (arrows) of human AMD retina, compared to the control (CTL). Sections are counterstained for 4′,6-diamidino-2-phenylindole (DAPI; blue), which labels cell nuclei. Images were obtained at 60×. Scale bar: 40 μm. BrM: Bruch’s membrane, GCL: ganglion cell layer, ONL: outer nuclear layer, PR: photoreceptor layer, RPE: retinal pigment epithelium.

**Figure 4 pharmaceuticals-13-00130-f004:**
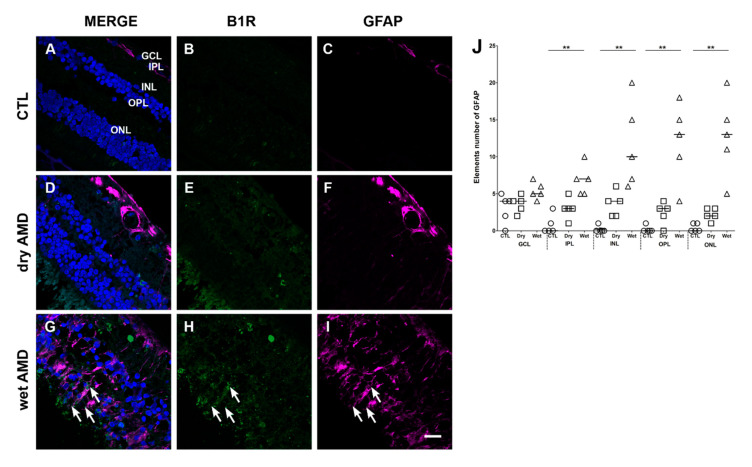
B1R colocalization with the glial fibrillary acidic protein (GFAP) macroglia marker in human retinae. (**A**–**I**) Microphotographs showing GFAP immunoreactivity and its co-expression with the B1R. GFAP immunoreactivity (purple) is co-expressed with B1R only in the wet AMD (**G**). Sections are all counterstained for 4′,6-diamidino-2-phenylindole (DAPI; blue), which labels cell nuclei (**A**,**D**,**G**). Images were obtained at 40×. Scale bar: 20 μm. (**J**) Semi-quantification of GFAP fluorescent projections on all the surface area of each retinal layer from five dry AMD, five wet AMD and five controls (CTL). GFAP was significantly increased (** *p* < 0.005) in all the layers, except GCL, of wet AMD retinae in comparison with the control. Data represent mean intensity ± SEM after subtraction of the background intensity. CTL: control (O), Dry: dry AMD (□), Wet: wet AMD (△), GCL: ganglion cell layer, IPL: inner plexiform layer, INL: inner nuclear layer, OPL: outer plexiform layer, ONL: outer nuclear layer.

**Figure 5 pharmaceuticals-13-00130-f005:**
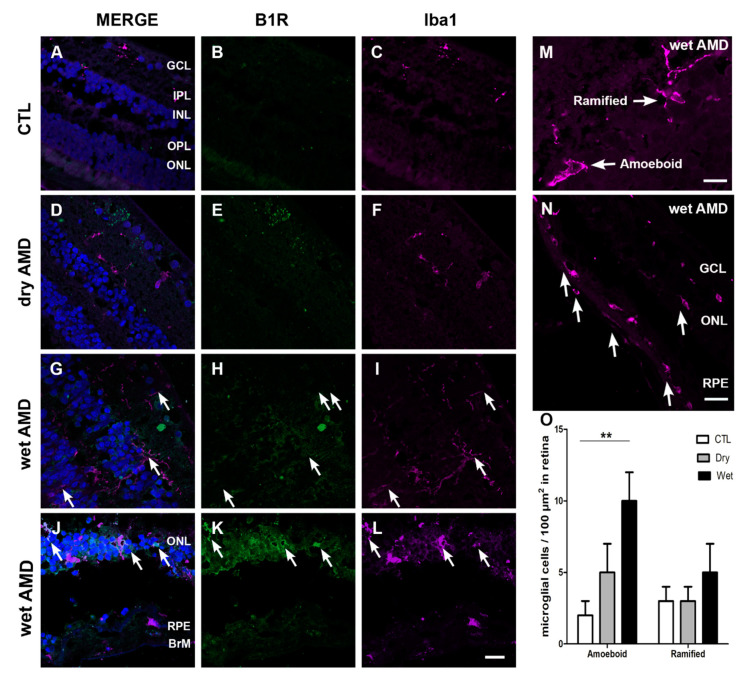
B1R colocalization with microglia in human retinae. (**A–L**) Microphotographs showing ionized calcium-binding adapter molecule 1 (Iba1) immunoreactivity and co-expression with B1R. Iba1 (purple) was markedly increased in all the layers of wet AMD retinae (arrows in **I** and **L**), when compared to the control (**C**) and dry AMD (**F**) retinae. In both forms of AMD, microglia activation was observed (arrow in **F**, **I** and **L**), indicated by hypertrophic somatic areas and highly activated microglia. Iba1 double immunostaining with B1R shows a co-expression of these two proteins all over the wet AMD retinae only (**G** and **J**). Sections are all counterstained for 4′,6-diamidino-2-phenylindole (DAPI; blue), which labels cell nuclei (**A**,**D**,**G**,**J**). (**M**) ramified and amoeboid shape microglia. (**N**) Infiltrating microglia in the RPE layer. Images are: **A**–**L**; 40× (20 μm), **M**; 60× (10 μm) **N**; 20× (40 μm). (**O**) Counting of ramified and amoeboid microglia in the retina. Data represent mean intensity ± SEM after subtraction of the background intensity. ** *p* < 0.005 wet AMD compared with the control (CTL). BrM: Bruch’s membrane, GCL: ganglion cell layer, IPL: inner plexiform layer, INL: inner nuclear layer, OPL: outer plexiform layer, ONL: outer nuclear layer, RPE: retinal pigment epithelium.

**Figure 6 pharmaceuticals-13-00130-f006:**
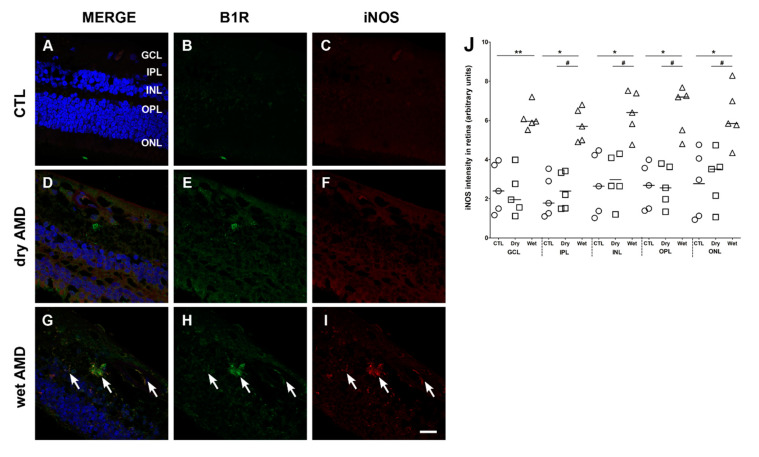
B1R colocalization with inducible nitric oxide synthase (iNOS) in human retinae. (**A–I**) Microphotographs showing iNOS immunoreactivity and its co-expression with the B1R. iNOS immunoreactivity (red) was slightly detectable in the dry AMD (**F**) and the control retinae (**C**), but highly expressed in all layers of the wet AMD retinae (arrows in **I**). A colocalization of iNOS with B1R was observed in all layers of the wet AMD (**G**), but not in the dry AMD (**D**) or control (**A**) retinae. Sections are all counterstained for 4′,6-diamidino-2-phenylindole (DAPI; blue), which labels cell nuclei (**A**,**D**,**G**). Images were obtained at 40×. Scale bar: 20 μm. (**J**) Semi-quantification of iNOS fluorescence intensity on all the surface area of each retina from five dry AMD, five wet AMD and five controls. Data represent mean intensity ± SEM after subtraction of the background intensity. * *p* < 0.05, ** *p* < 0.005 wet AMD compared with the control (CTL), ^#^
*p* < 0.005 dry AMD compared with the wet AMD. CTL: control (O), Dry: dry AMD (□), Wet: wet AMD (△), GCL: ganglion cell layer, IPL: inner plexiform layer, INL: inner nuclear layer, OPL: outer plexiform layer, ONL: outer nuclear layer.

**Figure 7 pharmaceuticals-13-00130-f007:**
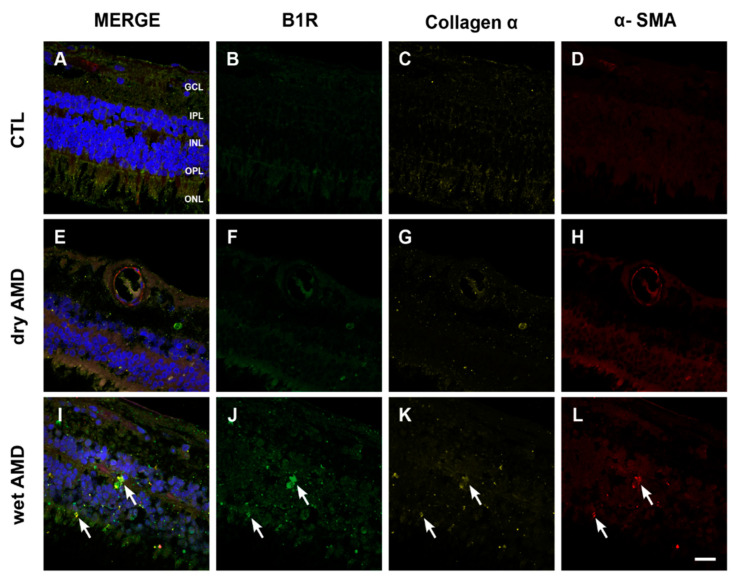
B1R colocalization with fibrosis markers in human retinae. (**A**–**L**) Microphotographs showing collagen 1α (yellow) and α-SMA (red) immunoreactivity and their co-expression with the B1R (green) in human retinae. Collagen 1α and α-SMA immunoreactivity was slightly detectable in the dry AMD ((**G**) and (**H**)) and control retinae ((**C**) and (**D**)), but strongly increased in the wet AMD retinae (arrows in K and L). A colocalization of collagen 1α and α-SMA with B1R was observed in the outer layers of the wet AMD (**I**) but not in the dry AMD (**E**) or control (**A**) retinae. Sections are all counterstained for 4′,6-diamidino-2-phenylindole (DAPI; blue), which labels cell nuclei (**A**,**E**,**I**). Images were obtained at 40×. Scale bar: 20 μm. GCL: ganglion cell layer, IPL: inner plexiform layer, INL: inner nuclear layer, OPL: outer plexiform layer, ONL: outer nuclear layer.

**Figure 8 pharmaceuticals-13-00130-f008:**
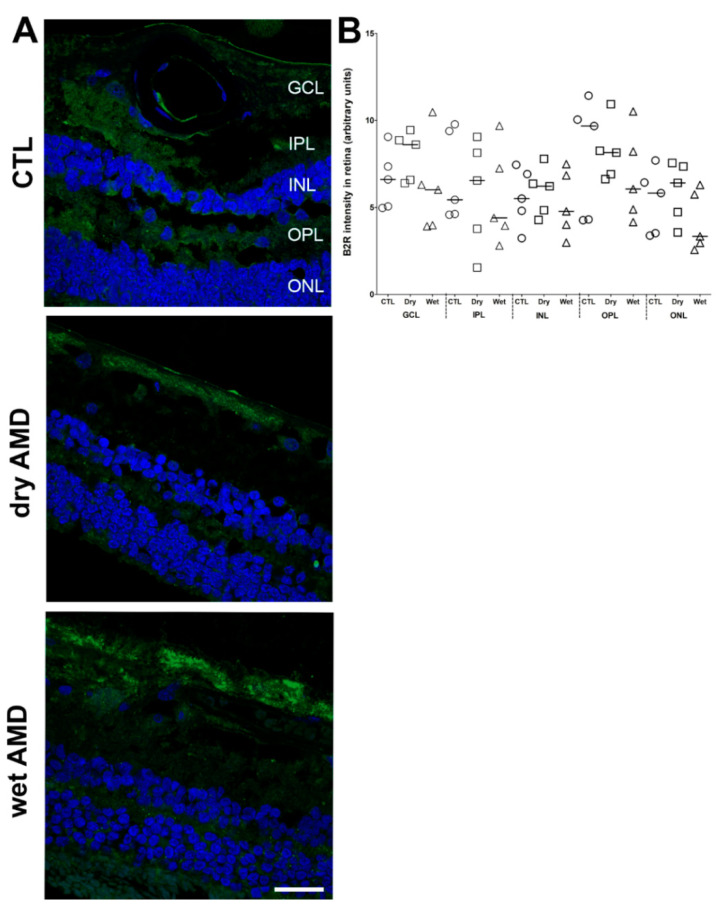
B2R immunoreactivity in human retinae. (**A**) Microphotographs of immunolocalization of B2R in the human control (CTL), dry and wet AMD retinae. All three groups express the same protein levels of B2R, particularly in the GCL and ONL. Sections are counterstained for 4′,6-diamidino-2-phenylindole (DAPI; blue), which labels cell nuclei. Images were obtained at 40×. Scale bar: 20 μm. (**B**) Semi-quantification of B2R fluorescence intensity on all the surface area of each retina from five dry AMD, five wet AMD and five control retinae. Data represent mean intensity ± SEM after subtraction of the background intensity. CTL: control (O), Dry: dry AMD (□), Wet: wet AMD (△), GCL: ganglion cell layer, IPL: inner plexiform layer, INL: inner nuclear layer, OPL: outer plexiform layer, ONL: outer nuclear layer.

**Table 1 pharmaceuticals-13-00130-t001:** Characteristics of retinae donors.

	Control		Dry AMD		Wet AMD	
Patients	Age	Sex	Age	Sex	Age	Sex
1	72	F	88	M	92	M
2	59	F	95	F	102	M
3	73	M	83	F	90	M
4	86	M	77	F	98	F
5	65	M	70	M	76	M
**Mean ± SEM**	71 ± 5		83 ± 4		92 ± 4	

F: Female, M: Male.

**Table 2 pharmaceuticals-13-00130-t002:** List of primary antibodies.

Antigen	Antibody Concentration and Source
**B1R**	Rabbit polyclonal, 1:100, Biotechnology Research Institute, Montreal, QC, Canada [67]
**B2R**	Rabbit polyclonal, 1:50, ABR-012, Alomone labs, Jerusalem, Israel
**iNOS**	Mouse monoclonal, 1:200, MAB9502, R&D systems, Oakville, ON, Canada
**Iba1**	Mouse monoclonal, 1:200, MABN92, EMD Millipore, Oakville, ON, Canada
**GFAP**	Chicken polyclonal, 1:1000, AB4674, Abcam, Cambridge, MA, USA
**Collagen 1α**	Goat polyclonal, 1:50, AF6220, R&D systems, Oakville, ON, Canada
**α-SMA**	Mouse monoclonal, 1:50, M0851, Dako, Burlington, ON, Canada

**Table 3 pharmaceuticals-13-00130-t003:** Summary of the differences in protein expression between the control, dry and wet AMD retinae.

Markers	Control	Dry AMD	Wet AMD
B1R	+/−	+	+++
GFAP	+	++	+++
Iba1	+ (ramified)	++ (Amoeboid> ramified)	+++ (Amoeboid > ramified)
iNOS	−	+/−	+++
Fibrosis	−	−	+++
B2R	+	+	+

+/−; slight or absence, +; low, ++ medium, +++ high immunoreactivity.

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
