# Peer review of "Differential Expression of Kinin Receptors in Human Wet and Dry Age-Related Macular Degeneration Retinae"

_pharmaceuticals, 2020, doi:10.3390/ph13060130_

Round 1
Reviewer 1 Report
Othman et al. realized a very interesting article analyzing the differential expression of kinin receptors in human wet and dry AMD retinae. I consider the manuscript structured in a very clear way but, in the same time, I suggest several revisions needed to improve the completeness of the paper, especially regarding references that I suggest to add:
- 1 sub-chapter: “RPE layer separates the retina from the choroid, providing a protection against outer insults. Hence, an alteration of this layer exposes the retina to the choroidal circulation and to an infiltration of pro-inflammatory immunomodulators”. This period lacks of references. I suggest to add the followings (PMID: 27372654 and PMID: 28764803), that describe how choroidal neovascularization in patients with AMD and retinitis pigmentosa (RP) induces the loss of the outer blood-retina barrier by down-expression of specific genes like RLBP1, RPE65 and VEGFA.
- 3 sub-chapter: the authors describe several molecular mechanisms depicting the role of oxidative stress in AMD. Recent literature has deepened such mechanisms and unveiled new ones, regarding non only coding genes but also non-coding. I suggest to add several references about these innovative molecular sides related to AMD and other retinal diseases, such as (PMID: 32290199, PMID: 32326576 and PMID: 29435412).
- 3 sub-section: the authors performed RNA Scope assay to visualize and quantify the genomic expression of their selected genes. Why did the authors use this technique rather than more performing one such as RNA-Sequencing?
- Even if the manuscript is well written, it requires minor English revisions and typos correction.
Author Response
Othman et al. realized a very interesting article analyzing the differential expression of kinin receptors in human wet and dry AMD retinae. I consider the manuscript structured in a very clear way but, in the same time, I suggest several revisions needed to improve the completeness of the paper, especially regarding references that I suggest to add:
- 1 sub-chapter: “RPE layer separates the retina from the choroid, providing a protection against outer insults. Hence, an alteration of this layer exposes the retina to the choroidal circulation and to an infiltration of pro-inflammatory immunomodulators”. This period lacks of references. I suggest to add the followings (PMID: 27372654 and PMID: 28764803), that describe how choroidal neovascularization in patients with AMD and retinitis pigmentosa (RP) induces the loss of the outer blood-retina barrier by down-expression of specific genes like RLBP1, RPE65 andVEGFA.
Reply: We added one reference suggested by the reviewer to illustrate this point (ref 40, lines 240-242).
- 3 sub-chapter: the authors describe several molecular mechanisms depicting the role of oxidative stress in AMD. Recent literature has deepened such mechanisms and unveiled new ones, regarding non only coding genes but also non-coding. I suggest to add several references about these innovative molecular sides related to AMD and other retinal diseases, such as (PMID: 32290199, PMID: 32326576 and PMID: 29435412).
Reply: We added the 3 references mentioned by the reviewer (refs 54-56, lines 293-295).
- 3 sub-section: the authors performed RNA Scope assay to visualize and quantify the genomic expression of their selected genes. Why did the authors use this technique rather than more performing one such as RNA-Sequencing?
Reply: It would be an interesting study to perform RNA-sequencing with human AMD retina to have a complete picture of the genes affected by this disease. However, the aim of our study was to quantify some selected gene on our fixed paraffin-embedded tissues. RNAscope provides a quantitative value of gene expression as well as a precise cellular localization of the genes that RNASEq does not provide.
- Even if the manuscript is well written, it requires minor English revisions and typos correction.
Reply: We revised the entire manuscript and made typos correction. See main changes in yellow colour in the text.
Thank you for your valuable comments and suggestions!
Reviewer 2 Report
The aim of the project was to investigate the expression and functions of the kinin B1 and B2 receptors (B1R and B2R) in human retina during age-related macular degeneration (AMD). The authors studied the retina from post-mortem human eyes obtained from AMD patients and they compared the data with the control. The data show a significant increase in B1R mRNA and immunoreactivity in AMD patients as compared with the control group, while no change was seen in B2R. The authors conclude that the kinin receptor B2R is important for the inflammatory response during AMD and it may be a target for preventing retinal inflammation in AMD.
The abstract is not very clear. I understand the word limitation, but at this stage it is not clear. In the first paragraph, the authors should specify the role of kinin receptors during inflammation. The aim is not clear: what does characterize mean? In the last sentence, what does “key pathological event” mean?
Do the authors have any hematoxilyn and eosin histology to show tissue damage?
Did the authors try to see if there was any sex-related changes?
Did the authors do any western blotting on the tissues?
The discussion is too long.
Author Response
The aim of the project was to investigate the expression and functions of the kinin B1 and B2 receptors (B1R and B2R) in human retina during age-related macular degeneration (AMD). The authors studied the retina from post-mortem human eyes obtained from AMD patients and they compared the data with the control. The data show a significant increase in B1R mRNA and immunoreactivity in AMD patients as compared with the control group, while no change was seen in B2R. The authors conclude that the kinin receptor B2R is important for the inflammatory response during AMD and it may be a target for preventing retinal inflammation in AMD.
The abstract is not very clear. I understand the word limitation, but at this stage it is not clear. In the first paragraph, the authors should specify the role of kinin receptors during inflammation. The aim is not clear: what does characterize mean? In the last sentence, what does “key pathological event” mean?
Reply: Please see the new version of the Abstract (yellow colour).
Do the authors have any hematoxilyn and eosin histology to show tissue damage?
Reply: No, we were limited in experiments due to the few number of retina sections available.
Did the authors try to see if there was any sex-related changes?
Reply: This is an interesting point that we were looking for. However, we could have only five eyes (without equal number of male and female, Table 1) in each group from the Eyebank, which is not sufficient to see if there is any correlation with sex.
Did the authors do any western blotting on the tissues?
Reply: We only had access to already fixed and mounted slides. Fresh retinae were not available to look at the protein expression in the whole retinal tissues by Western Blot.
The discussion is too long.
Reply: We revised the discussion, trying to be more concise. We removed few sentences and the paragraph 3.6 containing former references 68, 69. However, reviewer 1 asked to add some citations and better discuss some points. We also believe that it is worth to discuss each of the markers studied.
We thank you for your valuable comments and suggestions!
Round 2
Reviewer 1 Report
The authors addressed all suggested points.
Reviewer 2 Report
The authors addressed all the reviewers' concerns. Good luck with your future work!